# Analytical and Clinical Validation of the ConfiSign HIV Self-Test for Blood-Based HIV Screening

**DOI:** 10.3390/diagnostics15141833

**Published:** 2025-07-21

**Authors:** Hyeyoung Lee, Ae-Ran Choi, Hye-Sun Park, JoungOk Kim, Seo-A Park, Seungok Lee, Jaeeun Yoo, Ji Sang Yoon, Sang Il Kim, Yoon Hee Jun, Younjeong Kim, Yeon Jeong Jeong, Eun-Jee Oh

**Affiliations:** 1Department of Laboratory Medicine, International St. Mary’s Hospital, College of Medicine, Catholic Kwandong University, Incheon 22711, Republic of Korea; shomermaid@catholic.ac.kr; 2Department of Laboratory Medicine, Seoul St. Mary’s Hospital, College of Medicine, The Catholic University of Korea, Seoul 06591, Republic of Korea; bibi@cmcnu.or.kr (A.-R.C.); totoroskyhs@naver.com (H.-S.P.); 5708ksy@naver.com (J.K.); gary431@naver.com (J.S.Y.); 3Resesarch and Development Institute for In Vitro Diagnostic Medical Devices, College of Medicine, The Catholic University of Korea, Seoul 06591, Republic of Korea; inna9908@naver.com; 4Department of Medical Sciences, Graduate School, The Catholic University of Korea, Seoul 06591, Republic of Korea; 5Department of Laboratory Medicine, Incheon St. Mary’s Hospital, College of Medicine, The Catholic University of Korea, Incheon 21431, Republic of Korea; lsok@catholic.ac.kr (S.L.); focused@catholic.ac.kr (J.Y.); 6Division of Infectious Diseases, Department of Internal Medicine, Seoul St. Mary’s Hospital, College of Medicine, The Catholic University of Korea, Seoul 06591, Republic of Korea; drksi@catholic.ac.kr (S.I.K.); koreajunmd@gmail.com (Y.H.J.); 7Division of Infectious Diseases, Department of Internal Medicine, Incheon St. Mary’s Hospital, College of Medicine, The Catholic University of Korea, Incheon 21431, Republic of Korea; muze1004@catholic.ac.kr (Y.K.); onlyyjyj@naver.com (Y.J.J.)

**Keywords:** HIV, self-testing, blood based, screening

## Abstract

**Background/Objectives**: Since the World Health Organization (WHO) recommended HIV self-testing as an alternative to traditional facility-based testing in 2016, it has been increasingly adopted worldwide. This study aimed to evaluate the performance of the ConfiSign HIV Self-Test (GenBody Inc., Republic of Korea), a newly developed blood-based immunochromatographic assay for the qualitative detection of total antibodies (IgG and IgM) against HIV-1/HIV-2. **Methods**: The evaluation included four components: (1) retrospective analysis of 1400 archived serum samples (400 HIV-positive and 1000 HIV-negative samples), (2) prospective self-testing by 335 participants (112 HIV-positive participants and 223 individuals with an unknown HIV status, including healthy volunteers), (3) assessment using seroconversion panels and diverse HIV subtypes, and (4) analytical specificity testing for cross-reactivity and interference. The Elecsys HIV combi PT and Alinity I HIV Ag/Ab Combo assays were used as reference assays. **Results**: In retrospective testing, the ConfiSign HIV Self-Test achieved a positive percent agreement (PPA) of 100%, a negative percent agreement (NPA) of 99.2%, and a Cohen’s kappa value of 0.986, showing excellent agreement with the reference assays. In the prospective study, the test showed 100% sensitivity and specificity, with a low invalid result rate of 1.8%. All HIV-positive samples, including those with low signal-to-cutoff (S/Co) values in the Alinity I assay, were correctly identified. The test also reliably detected early seroconversion samples and accurately identified a broad range of HIV-1 subtypes (A, B, C, D, F, G, CRF01_AE, CRF02_AG, and group O) as well as HIV-2. No cross-reactivity or interference was observed with samples that were positive for hepatitis viruses, cytomegalovirus, Epstein–Barr virus, varicella zoster virus, influenza, HTLV-1, HTLV-2, or malaria. **Conclusions**: The ConfiSign HIV Self-Test demonstrated excellent sensitivity, specificity, and robustness across diverse clinical samples, supporting its reliability and practicality as a self-testing option for HIV-1/2 antibody detection.

## 1. Introduction

HIV/AIDS continues to be a significant global public health concern, with approximately 39.9 million people living with HIV worldwide by the end of 2023 [1]. Despite progress in treatment and prevention, 1.3 million people acquired new HIV infections during 2023, and an estimated 630,000 people died from AIDS-related causes [1].

A significant advancement in HIV diagnosis is the development of rapid HIV tests, which have enabled HIV screening to be conducted outside of traditional clinical laboratories. These tests are broadly categorized into point-of-care tests, performed by healthcare professionals near the patient, and self-tests, which are designed for individuals to use independently at home or in other non-clinical settings [2]. HIV self-testing (HIVST) overcomes several barriers associated with traditional HIV testing by ensuring privacy, minimizing stigma, and offering a convenient alternative that does not require visits to healthcare facilities. Since the World Health Organization (WHO) first recommended HIVST as an alternative to traditional facility-based testing in 2016 [3], there has been rapid global adoption of this approach [4]. This expansion builds on UNITAID’s 2018 findings, which emphasized the importance of HIVST for key and underserved populations who might not otherwise access facility-based testing [5]. As a result, more than 10 million kits are now distributed annually, and nearly 100 countries have incorporated HIVST into their national programs [6].

However, current HIV self-testing options still face challenges, such as the variable sensitivity in early infection, higher rates of invalid results, higher costs, and limited accessibility or usability among certain populations, highlighting the need for new, reliable, and user-friendly self-testing devices.

The ConfiSign HIV Self-Test (Genbody, Cheonan-si, Republic of Korea) is a newly developed blood-based rapid test specifically designed for self-testing. It detects a broad range of antibodies against HIV-1 subtypes and HIV-2 using a small blood sample and provides results through a simple visual readout, making it accessible for non-professional users.

In this study, we comprehensively evaluated the clinical performance of the ConfiSign HIV Self-Test by comparing its results with standard laboratory reference methods, utilizing both retrospective clinical serum samples and prospective self-testing by participants. Beyond standard clinical accuracy, we also assessed the test’s performance in detecting early seroconversion, its efficacy across a broad spectrum of HIV subtypes, and its analytical specificity, including cross-reactivity and interference with other infections or endogenous substances.

## 2. Materials and Methods

### 2.1. Study Design and Assay Principle

This study consisted of four main evaluations of the ConfiSign HIV Self-Test: (1) assessment of performance using retrospective leftover clinical samples, (2) prospective evaluation involving self-testing, (3) analysis of performance using seroconversion and subtype-specific panels, and (4) analytical specificity evaluation including assessment across different specimen types and cross-reactivity analysis. The ConfiSign HIV Self-Test utilizes a nitrocellulose membrane with immobilized HIV-1 antigens (gp41, gp120) and HIV-2 antigen (gp36) on the test line. Recombinant HIV antigens are conjugated to colloidal gold particles and placed on a polyester pad within the device. When the blood sample is taken into the test device, the solubilized conjugate migrates with the sample by passive diffusion, allowing both the conjugate and sample to interact with the recombinant antigens on the nitrocellulose. If the sample contains antibodies against HIV-1 or HIV-2, the visible red line appears within 15 min. The solution continues to migrate to the control region, where it binds a control conjugate, producing another red line that confirms the correct test operation.

### 2.2. Retrospective Clinical Performance Evaluation

A total of 1400 serum samples were obtained from leftover specimens collected at Seoul St. Mary’s Hospital, Republic of Korea. The HIV-positive serum samples (*n* = 400) and HIV-negative serum samples (*n* = 1000) were originally obtained for routine HIV screening or diagnostic testing and analyzed using an electrochemiluminescent immunoassay (ECLIA) with the Elecsys HIV combi PT system (Roche Diagnostics GmbH, Penzberg, Germany) or a chemiluminescent microparticle immunoassay (CMIA)-based HIV-1/HIV-2 combination antigen/ antibody (Ag/Ab) test CM (Abbott Alinity HIV Ag/Ab Combo test, Abbott Laboratories, Abbott Park, IL, USA). Blood specimens were included in the clinical trial if they were collected from participants who met all of the eligibility criteria, including confirmed HIV status by immunoassays, proper anonymization, and adherence to standardized collection and storage protocols. Specimens were excluded if they exhibited signs of microbial contamination, exhibited marked hemolysis or turbidity, were subject to improper storage or unverifiable storage conditions, were pooled from multiple sources, or had a volume less than 100 µL. All specimens were stored at −80 °C and were thawed and centrifuged prior to testing with the ConfiSign HIV Self-Test. The study protocol was reviewed and approved by the Institutional Review Board of Seoul St. Mary’s Hospital (KC23DOSV0535).

### 2.3. Prospective Clinical Performance Evaluation Study

The prospective study was conducted from 1 October 2024 to 31 December 2024 at two tertiary care hospitals—Seoul St. Mary’s Hospital and Incheon St. Mary’s Hospital (Republic of Korea). Individuals aged 13 years or older who voluntarily agreed to participate and provided written informed consent were eligible for enrollment. All participants were required to be capable of independently collecting capillary blood samples and interpreting the test results. For minors under 19 years of age, both the participant and their legal guardian received a full explanation of the study and co-signed the informed consent form. Although the study protocol allowed for the inclusion of participants aged 13 to 18 years with guardian consent, no minors were actually enrolled in the study. HIV-positive participants were individuals diagnosed with HIV/AIDS, as confirmed by medical records, who visited the study institutions. HIV-negative participants were individuals with no known risk factors or symptoms of HIV infection and who were unaware of their HIV status. A total of 112 HIV-positive individuals with a confirmed diagnosis and 223 individuals with an unknown HIV status, including healthy volunteers were recruited. Exclusion criteria included being aged under 13 years, a failure to provide written informed consent, or psychological instability as determined by the principal investigator. Samples were also excluded if consent was withdrawn, if there was evidence of sample contamination or insufficient sample volume, or if documentation was incomplete. Participants were provided with the manufacturer’s instructions and performed the ConfiSign HIV Self-Test independently using fingertip capillary blood. Results were self-reported as positive, negative, or invalid. Invalid results were retested once with approval by the study staff. Additionally, 5 mL of venous blood was drawn by medical staff and tested using the Abbott Alinity HIV Ag/Ab Combo assay as the reference method. This study protocol was reviewed and approved by the Institutional Review Board (IRB No. XC24DDDV0061).

### 2.4. Seroconversion Panels and Genotype-Specific Samples

A total of 31 commercially available HIV seroconversion panels, representing 298 samples, were tested. Of these, 23 panels comprising 250 specimens were purchased from ZeptoMetrix (Buffalo, NY, USA): HIV12007, HIV12008, HIV6248, HIV9011, HIV9014, HIV9018, HIV9019, HIV9020, HIV9021, HIV9025, HIV9026, HIV9030, HIV9031, HIV9033, HIV9034, HIV9075, HIV9076, HIV9077, HIV9079, HIV9082, HIV9084, HIV9089, and HIV9096, and 8 panels comprising 48 specimens, were obtained from SeraCare Inc. (Milford, MA, USA): AccuVert HIV-1 Seroconversion Panel (0600-0272), AccuVert HIV-1 Seroconversion Panel (0600-0251), AccuVert HIV-1 Seroconversion Panel (0600-0249), AccuVert HIV-1 Seroconversion Panel (0600-0258), AccuVert HIV-1 Seroconversion Panel (0600-0270), AccuVert HIV-1 Seroconversion Panel (0600-0248), AccuVert HIV-1 Seroconversion Panel (0600-0262), and AccuVert HIV-1 Seroconversion Panel (0600-0252).

To assess strain inclusivity, 72 archived plasma samples representing a range of HIV subtypes were purchased from Biomex GmbH (Heidelberg, Germany). This set included confirmed HIV-1 samples from several group M subtypes (such as A, B, C, D, and circulating recombinant forms like CRF01_AE and CRF02_AG) as well as group O samples. All panel specimens had been previously characterized using reference laboratory methods to conform their HIV status and subtype. The HIV-2 subset included 12 HIV-2 Western-blot-positive plasma specimens provided by the Ministry of Food and Drug Safety (MFDS, Osong, Republic of Korea).

### 2.5. Analytical Specificity Across Specimen Types and Cross-Reactivity Analysis

To compare performance across various specimen types, samples from 10 HIV patients who were confirmed to be HIV-1 antibody positive were collected and tested using fingerstick whole blood, sera, plasma, and venous whole blood. This prospective study was approved by the IRB of Seoul St. Mary’s Hospital (KC23DSSV0616), and all participants provided written informed consent.

Cross-reactivity and interference studies were also conducted. A total of 66 HIV-negative serum samples known to be positive for other infectious agents, including hepatitis A, B, and C viruses; cytomegalovirus (CMV); Epstein–Barr virus (EBV); varicella zoster virus (VZV); influenza A and B viruses; human T-lymphotropic virus types 1 and 2 (HTLV-1/2); and malaria, were evaluated. In addition, the potential for interference from common endogenous substances was assessed using HIV-negative serum samples spiked with high levels of bilirubin (icterus), free hemoglobin (hemolysis), and lipids (lipemia). All samples were tested according to the manufacturer’s instructions.

### 2.6. Statistical Analysis

Diagnostic performance was assessed by calculating the sensitivity, specificity, agreement rate, Cohen’s kappa coefficient, positive percent agreement (PPA), and negative percent agreement (NPA), each with corresponding 95% confidence intervals (CIs). PPA and NPA were calculated by comparing the results of the ConfiSign HIV Self-Test with those obtained from immunoassays used as reference standards. Confidence intervals for proportions were calculated using the Clopper–Pearson (exact) method. Cohen’s kappa coefficient was calculated to assess the level of agreement beyond chance. All statistical analyses were performed using MedCalc^®^ Statistical Software version 23.2.8 (MedCalc Software Ltd., Ostend, Belgium) and GraphPad Prism version 10.5.0 (GraphPad Software, San Diego, CA, USA). All statistical tests were two tailed, with *p*-values of <0.05 being considered statistically significant. For the retrospective study, the sample size was based on the requirements set by the Ministry of Food and Drug Safety (MFDS) of Korea for approval of HIV immunoassay devices, which recommend 400 HIV-positive samples and 1000 HIV-negative samples. For the prospective study, the sample size was determined based on the maximum number of eligible HIV-positive individuals who could be enrolled during the study period across two tertiary hospitals.

## 3. Results

### 3.1. Clinical Performance Using Leftover Serum Samples

A total of 1400 leftover clinical serum samples, comprising 400 HIV-positive and 1000 HIV-negative samples as determined by reference laboratory assays, were evaluated. The ConfiSign HIV Self-Test correctly identified all 400 HIV-positive samples, yielding a PPA of 100.0% (95% CI: 99.1–100.0%). Of the 1000 negative specimens, 8 (0.8%) were incorrectly identified as positive resulting in an NPA of 99.2% (95% CI: 98.4–99.7%) (Table 1). The corresponding sensitivity, specificity, PPV, and NPV were 100.0% (95% CI: 99.0–100.0%), 99.2% (95% CI: 98.4–99.7%), 98.0% (95% CI: 96.2–99.0%), and 100.0% (95% CI: 99.6–100.0%), respectively. Cohen’s kappa coefficient was 0.986, indicating excellent concordance with the reference method.

### 3.2. Clinical Performance in Prospective Self-Testing

A total of 335 participants were enrolled including 112 HIV-positive and 223 HIV-negative individuals, of whom 172 (51.3%) were male and 163 (48.7%) were female. The median age of participants was 37.0 years (interquartile range: 31.0–47.0 years). Among the 335 participants, 96.4% of the HIV-positive group were male compared to 28.7% in the HIV-negative group. The HIV-positive group also had a higher proportion of individuals aged 50 years and older (42.0% vs. 11.6%), while all participants were of Asian ethnicity. The ConfiSign HIV Self-Test demonstrated 100% sensitivity and specificity (95% CI: 96.8–100% and 98.4–100%, respectively), and 100% PPV and NPV (95% CI: 96.8–100% and 98.4–100%, respectively), correctly identifying all HIV-positive and HIV-negative samples. Six participants (1.8%) obtained invalid results on the initial attempt but, obtained valid negative results upon retesting. The median signal-to-cutoff (s/co) value measured by the Alinity assay for HIV-positive samples was 336.9 (95% CI: 334.5–420.4), while, HIV-negative samples had a median s/co of 0.06 (95% CI: 0.05–0.06), ensuring clear differentiation between positive and negative results (Figure 1). Notably, the ConfiSign HIV Self-Test successfully detected HIV-positive cases even among samples with relatively low s/co values (1–10 S/Co), indicating reliable performance across a range of antibody concentrations.

### 3.3. Performance in Seroconversion and Diverse HIV Subtypes

A total of 31 HIV seroconversion panels, comprising 298 samples, were evaluated to assess the early detection capability of the ConfiSign HIV Self-Test (Table 2). The ConfiSign HIV Self-Test detected 75 positive samples (25.7%), compared to 96 positives (32.2%) reported in the manufacturer’s insert using the 4th generation HIV Ag/Ab combo assay. The ConfiSign HIV Self-Test demonstrated identical detection performance to the reference method in 14 out of the 31 panels, with minor differences in the remaining panels. While the ConfiSign HIV Self-Test detected slightly fewer early seroconversion samples than the reference assay, it demonstrated reasonable performance for the early detection of HIV infection. The test also demonstrated broad reactivity across major HIV-1 subtypes (A, B, C, D, F, G, CRF01_AE, CRF02_AG) and successfully detected group O samples, supporting inclusivity for diverse HIV strains. All 12 HIV-2 positive samples were also correctly identified.

### 3.4. Performance Across Specimen Types and Cross-Reactivity Analysis

A subset of different specimen types, including fingerstick whole blood, sera, plasma, and venous whole blood from 10 HIV-positive patients, was tested. The ConfiSign HIV Self-Test demonstrated 100% concordance across all specimen types, confirming robust performance regardless of the sample matrix. None of the 66 hepatitis A/B/C, cytomegalovirus, Epstein–Barr virus, varicella zoster virus, influenza A/B, HTLV-1, HTLV-2, and malaria-positive samples tested showed cross-reactivity with the ConfiSign HIV Self-Test. Additionally, no interference was found in samples with icterus, hemolysis or lipemia.

## 4. Discussion

The ConfiSign HIV Self-Test demonstrated excellent clinical performance in this comprehensive clinical evaluation, suggesting its potential as a reliable self-testing tool for HIV. Under both retrospective and prospective analyses, ConfiSign HIV Self-Test showed 100% PPV, correctly identifying all HIV-positive samples. Its robust performance across a wide range of HIV-1 subtypes, successful detection of HIV-2, and consistent results with various specimen types suggest its potential for broad global applicability. The low invalidity rate (1.8%) further supports its practicality for self-testing in diverse settings.

The ConfiSign HIV Self-Test demonstrated 100% sensitivity and specificity in the prospective self-testing analysis. When compared to other commercially available HIV self-tests, the ConfiSign HIV Self-Test performed at least as well as, or better than them. A study conducted in South Africa evaluated the performance of three blood-based HIV self-test kits produced by BioSure Ltd. (Waltham Abbey, UK), bioLytical Laboratories (INSTI) (Richmond, BC, Canada), and Chembio Diagnostic Systems (Medford, NY, USA), as well as one oral-fluid-based kit produced by OraSure Technologies (Bethlehem, PA, USA) [7]. They reported that the sensitivity and specificity were 99.7% and 100% for BioSURE, 99% and 100% for INSTI, 96.8% and 100% for Chembio, and 99.3% and 99.4% for Orasure, respectively. HIVST can be classified by specimen type—either blood or oral fluid—with both methods providing results within 15 to 20 min [8]. A systematic review and meta-analysis found that blood-based HIVST kits have a higher sensitivity than oral-fluid kits, largely due to the higher antibody concentrations in blood and fewer user errors during sample collection [9]. A study conducted in Zambia evaluated the OraQuick HIV self-test against a fourth-generation laboratory reference test, reporting a sensitivity of 87.5% and a specificity of 99.7% [10]. Another cross-sectional study in Vietnam compared four blood-based HIV self-test kits (INSTI, SURE CHECK, BioSURE, and CheckNOW) to the gold-standard fourth-generation ELISA [11]. The combined sensitivity and specificity of these kits were 96.4% and 99.9%, respectively. Individually, the sensitivities and specificities were as follows: INSTI: sensitivity 93.1%, specificity 99.8%; SURE CHECK: sensitivity 97.7%, specificity 100%; BioSURE: sensitivity 96.3%, specificity 100%; and CheckNOW: sensitivity 100%, specificity 99.8%.

Although the ConfiSign HIV Self-Test demonstrated high overall diagnostic accuracy, its lower sensitivity in early seroconversion cases should be carefully considered. As an antibody-only assay, it may yield false-negative results if used during the serological window period, when anti-HIV antibodies are not yet detectable in blood. This window period typically ranges from 2 to 12 weeks after potential exposure, depending on the individual’s immune response and the test characteristics [8,12,13]. Consequently, individuals who test negative but have had a recent high-risk exposure should be advised that a negative result does not exclude acute HIV infection. In such cases, repeat testing after the window period or the use of a fourth-generation antigen/antibody assay or nucleic acid amplification test is recommended to improve early detection [8]. Clear guidance regarding the optimal timing of self-testing and the limitations of antibody-only tests is essential to support informed decision-making and reduce the risk of missed early infections.

The study also included a broad range of HIV-1 subtypes (A, B, C, D, F, G, CRF01_AE, CRF02_AG) and group O, as well as HIV-2, confirming the test’s broad inclusivity and relevance for global use. While the ConfiSign HIV Self-Test demonstrated broad reactivity across major HIV-1 subtypes and HIV-2, our study included only a limited number of samples from certain rare subtypes and from HIV-2 infections. Although our preliminary findings suggest that the test is capable of detecting these variants, the small sample sizes do not allow for definitive conclusions regarding its performance in these groups. Therefore, additional validation studies in regions where these subtypes are more prevalent—such as West and Central Africa—are warranted. Such studies are essential to ensure the test’s diagnostic reliability and global applicability across diverse epidemiological settings. No cross-reactivity was observed with samples positive for other common infectious agents, nor was there interference from endogenous substances such as bilirubin, hemoglobin, or lipids, further underscoring the test’s specificity and reliability.

The test’s robust performance was further confirmed across various specimen types—including fingerstick whole blood, serum, plasma, and venous whole blood—demonstrating 100% concordance regardless of the sample matrix. This versatility is crucial for practical application in diverse settings, ensuring that users can obtain reliable results with different types of blood samples.

One concern with blood-based HIVST is the potential for user discomfort or procedural errors due to the finger-prick requirement [14]. However, our findings indicate that most participants were able to perform the test independently and interpret the results correctly, with a low rate of invalid outcomes (0.8%). According to a meta-analysis of HIV self-test kits, studies reported a higher proportion of invalid results when using blood-based tests (0.4–9.5%) compared to studies using oral-fluid-based rapid tests (0.2–4.5%). The observed user errors commonly described in studies included inadequate or incorrect specimen collection (such as improper fingerstick or oral swab techniques), incorrect or insufficient use of the buffer solution, improper transfer of blood samples, and difficulties in interpreting the test results [9]. While blood-based tests may have slightly higher invalidity rates than oral-fluid-based tests, the low rate of invalid outcomes demonstrated that performing the blood-based HIVST without external assistance was highly feasible. These findings are consistent with previous studies evaluating blood-based HIVST [11,15,16,17]. Well-illustrated instructions and recommendations for adequate lighting further support correct test usage.

This study has several limitations. First, the evaluation was conducted under controlled hospital settings, which may not fully reflect real-world conditions where self-testing is performed by the general population without supervision. Studies have shown that unassisted self-testers may commit procedural errors that could negatively impact the test performance. Second, the reduced sensitivity observed during early seroconversion in HIV self-tests is a well-documented limitation of antibody-based assays. For instance, a study evaluating the INSTI™ HIV-1 Antibody Test reported a sensitivity of 69.4% in detecting early HIV infections, highlighting the challenges that these tests face during the seroconversion window [18]. These findings underscore the importance of understanding the window period associated with antibody-based HIV tests and the need for confirmatory testing if recent exposure is suspected. Third, while we evaluated a broad range of HIV-1 subtypes and included group O samples, the sample size for certain rare subtypes and HIV-2 infections was limited. Although no sensitivity issues were observed in this study, additional validation in regions where HIV-2 or uncommon subtypes are more prevalent would strengthen the generalizability of our findings. Fourth, a notable limitation is the lack of detailed information on HIV risk profiles and pre-exposure prophylaxis (PrEP) use among participants in the HIV-negative cohort. As data on behavioral risk factors and PrEP status were not collected, it is difficult to determine the proportion of individuals at a higher risk of HIV acquisition or to assess the test performance among those on PrEP, who may experience delayed seroconversion. Future studies should incorporate risk stratification and PrEP usage data to enable more-accurate and population-specific assessments of HIV self-testing performance. Finally, detailed clinical information such as comorbidities or behavioral risk factors was not collected in this study.

In summary, the ConfiSign HIV Self-Test is a highly accurate, reliable, and practical tool for self-testing, with demonstrated strengths in analytical performance, usability, and inclusivity. Future research should focus on large-scale field studies in diverse populations and unsupervised settings to further assess the real-world effectiveness, user experience, and long-term impact on HIV diagnosis and care.

## 5. Conclusions

This comprehensive evaluation demonstrated that the ConfiSign HIV Self-Test offers excellent diagnostic performance, with high sensitivity, high specificity, and strong concordance with established laboratory assays. Its ability to accurately detect HIV-1/2 antibodies across diverse subtypes, including seroconversion samples, further underscores its diagnostic reliability. The absence of cross-reactivity with other pathogen infections supports its analytical specificity. With a low rate of invalid results and proven usability in self-testing settings, the ConfiSign HIV Self-Test supports broader access to HIV testing.

## Figures and Tables

**Figure 1 diagnostics-15-01833-f001:**
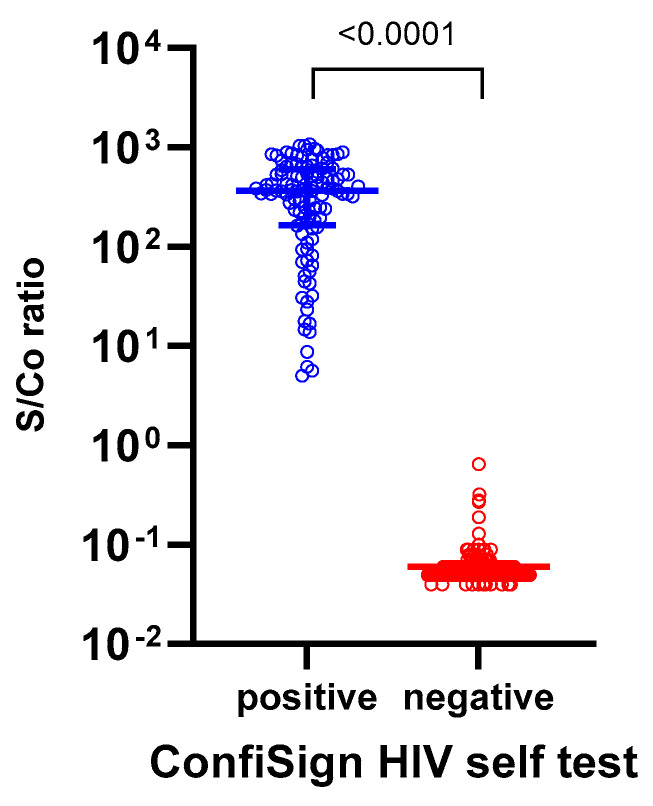
The signal-to-cutoff (S/Co) values measured by the Abbott Alinity assay among 112 HIV-positive and 223 HIV-negative participants in the prospective ConfiSign HIV-self testing study (*n* = 335). The S/Co value represents the ratio of the sample signal to the cutoff value established by the assay, with values ≥1.0 considered reactive (positive) and <1.0 considered non-reactive (negative).

**Table 1 diagnostics-15-01833-t001:** Positive and negative percent agreement of ConfiSign HIV Self-Test with laboratory reference assay based on retrospective testing of 1400 samples and prospective self-testing of 335 samples.

Performance of ConfiSign HIV Self-Test	Retrospective, 1400 Samples	Prospective, 335 Self-Testing Samples
True positive	400 (100%)	112 (100%)
False positive	8 (0.8%)	0 (0%)
True negative	992 (99.2%)	223 (100%)
False negative	0 (0%)	0 (0%)
Positive % agreement	100% (95% CI: 99.1–100.0%)	100% (95% CI: 96.8–100.0%)
Negative % agreement	99.2% (95% CI: 98.4–99.7%)	100% (95% CI: 98.4–100.0%)
Sensitivity %	100% (95% CI: 99.0–100.0%)	100% (95% CI: 96.8–100.0%)
Specificity %	99.2% (95% CI: 98.4–99.7%)	100% (95% CI: 98.4–100.0%)
Positive predictive value %	98.0% (95% CI: 96.2–99.0%)	100% (95% CI: 96.8–100.0%)
Negative predictive value %	100% (95% CI: 99.6–100.0%)	100% (95% CI: 98.4–100.0%)

**Table 2 diagnostics-15-01833-t002:** Analytical sensitivity of ConfiSign HIV Self-Test using seroconversion panel samples.

Seroconversion Panel	ConfiSign HIV Self-Test (Positive/Total)	Reference Method Result (Positive/Total)
HIV12007	5/9	6/9
HIV12008	4/13	5/13
HIV6248	1/7	2/7
HIV9011	2/11	2/11
HIV9014	4/5	4/5
HIV9018	2/10	3/10
HIV9019	1/3	1/3
HIV9020	2/22	3/22
HIV9021	2/17	4/17
HIV9025	1/12	2/12
HIV9026	1/7	1/7
HIV9030	2/16	3/16
HIV9031	3/19	3/19
HIV9033	2/16	2/16
HIV9034	1/13	3/13
HIV9075	1/4	2/4
HIV9077	11/23	12/23
HIV9076	3/9	3/9
HIV9079	5/15	7/15
HIV9082	1/4	1/4 *
HIV9084	1/4	1/4 *
HIV9089	1/5	2/5
HIV9096	2/6	2/6
0600-0272	2/6	3/6
0600-0251	3/10	3/10
0600-0249	3/6	3/6
0600-0270	2/4	2/4
0600-0252	2/4	4/4
0600-0262	2/4	2/4
0600-0248	2/10	3/10
0600-0258	1/4	2/4
Total	75/298 (25.7%)	96/298 (32.2%)

* Two panels (HIV9082 and HIV9084) were tested using the Abbott HIV Ab 1/2 EIA rDNA (3rd generation) assay.

## Data Availability

The data presented in this paper are available on request from the corresponding author.

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
