# Peer review of "Analytical and Clinical Validation of the ConfiSign HIV Self-Test for Blood-Based HIV Screening"

_diagnostics, 2025, doi:10.3390/diagnostics15141833_

Round 1
Reviewer 1 Report
Comments and Suggestions for Authors
This is a straightforwardstudy which clinically validated the ConfiSign HIV Self-2 Test for Blood-Based HIV Screening. The study has been appropriately designed and carried out. Please consider the following at the time of revising the maunscript:
- Page 2, Line 52-56: In introduction section, the HIV statistics given is 2023 and may be updated with 2024 data.
- Page 3, Line 122: The study included 13 years old in participating in the study. I understand that 13-year-old participants are considered as major as per the local legal procedures. If not, consent from the parents or guardians might be required.
The manuscript may be accepted for the publication after the revision/clarification based on the above queries.
Author Response
We sincerely thank the reviewer for the thorough and insightful comments, which have greatly helped improve the manuscript. We have addressed each point as follows:
Page 2, Line 52-56: In introduction section, the HIV statistics given is 2023 and may be updated with 2024 data.
- Thank you for your comment. However, as of July 2025, the official global HIV/AIDS statistics for 2024 have not yet been released by UNAIDS. The latest available data are from the end of 2023, as reported in the UNAIDS Global HIV & AIDS Statistics.
Page 3, Line 122: The study included 13 years old in participating in the study. I understand that 13-year-old participants are considered as major as per the local legal procedures. If not, consent from the parents or guardians might be required.
- Thank you for your insightful comment regarding the inclusion of participants aged 13 years and older (Page 3, Line 122). As per our IRB-approved protocol, participants under the age of 19 (legal minors) were required to be accompanied by a legal guardian, and both the participant and the guardian were required to sign the informed consent form. However, we would like to clarify that, in practice, no participants under the age of 19 were enrolled in the study.
- We have added this clarification to the revised manuscript in line 124-130 as “Individuals aged 13 years or older who voluntarily agreed to participate and provided written informed consent were eligible for enrollment. All participants were required to be capable of independently collecting capillary blood samples and interpreting test results. For minors under 19 years of age, both the participant and their legal guardian received a full explanation of the study and co-signed the informed consent form. Although the study protocol allowed for the inclusion of participants aged 13 to 18 years with guardian consent, no minors were actually enrolled in the study.”
Reviewer 2 Report
Comments and Suggestions for Authors
Major Issues:
1. Inconsistent Outcome Measures Between Cohorts
The retrospective cohort uses positive percent agreement (PPA), negative percent agreement (NPA), and Cohen’s kappa, while the prospective cohort uses sensitivity, specificity, and the invalid result rate. This inconsistency makes it difficult to directly compare the performance of the ConfiSign HIV Self-Test across the two cohorts. The authors should justify the use of different outcome measures or use the same set of indicators for both cohorts to ensure comparability.
2. Lack of Inclusion and Exclusion Criteria
The authors do not specify the inclusion and exclusion criteria for participants in either the retrospective or prospective cohorts. This information is critical for understanding the generalizability of the study findings and the potential biases that may have been introduced. For example, were participants with certain comorbidities excluded? Were there any restrictions based on the time since HIV diagnosis? The authors should provide detailed inclusion and exclusion criteria for both cohorts.
3. Insufficient Description of Statistical Methods
The statistical analysis section is too brief and lacks important details. For example, the authors mention calculating sensitivity, specificity, and Cohen’s kappa but do not describe how confidence intervals were computed or which statistical software was used. They also do not explain how sample sizes were determined for the retrospective and prospective studies. A more comprehensive description of the statistical methods is needed to allow readers to assess the validity of the results.
4. Limited Discussion of Early Seroconversion Performance
While the authors acknowledge that the ConfiSign HIV Self-Test has lower sensitivity in early seroconversion samples compared to fourth-generation assays, they do not discuss the implications of this limitation in sufficient detail. How might this affect the test’s real-world effectiveness? What guidance should be given to users regarding the timing of testing after potential exposure? The discussion should address these questions to provide a more balanced interpretation of the findings.
5. Generalizability of Subtype Inclusivity Findings
Although the test demonstrated broad reactivity across major HIV-1 subtypes and HIV-2, the sample size for certain rare subtypes and HIV-2 infections was limited. The authors should discuss whether the observed performance in these subtypes is sufficient to support global use of the test, particularly in regions where these subtypes are more common. Additional validation in diverse geographical settings may be necessary.
Minor Issues:
1. Clarity of Language
The English language needs further improvement for clarity and precision. For example:
Line 105-106: “...were originally submitted for routine HIV screening or diagnostic testing and analyzed using...” Here, “submitted” might be better replaced with “collected” or “obtained” to more accurately describe the process of sample acquisition.
Line 116: “The prospective study enrolled consenting adults between October 2024 and December 2024...” It would be clearer to specify the exact dates (e.g., “from October 1, 2024, to December 31, 2024”).
Line 121: “The target sample size included 112 HIV-positive individuals and 223 healthy volunteers with unknown HIV status.” The term “healthy volunteers with unknown HIV status” is contradictory. It would be better to state “223 individuals with unknown HIV status, including healthy volunteers.”
2. Terminology Consistency
The authors should ensure consistent use of terminology throughout the manuscript. For example, they use both “ConfiSign HIV Self-Test” and “Confisign HIV Self-Test” (e.g., Line 205). A single consistent spelling should be used.
3. Presentation of Results
The presentation of results in tables and figures could be improved. For example, in Table 1, the confidence intervals for PPA and NPA in the prospective study are not formatted consistently (e.g., “96.76%–100%” vs. “98.36%–100%”). Additionally, the figure caption for Figure 1 does not clearly explain what the S/Co ratio represents and how it was calculated. A more detailed caption would enhance understanding.
4. Missing References
Some statements lack appropriate references. For example, in the introduction (Line 56), the authors mention that “AIDS-related illnesses still claimed around 630,000 lives globally in 2023” but do not provide a reference for this statistic. All factual statements should be supported by references.
5. Formatting of Abbreviations
The list of abbreviations should be checked for consistency and completeness. For example, “HIV” and “AIDS” are defined, but terms like “PPA” and “NPA” are not included in the abbreviations section, even though they are used extensively in the text. Including all non-standard abbreviations in the list would improve readability.
Author Response
We sincerely thank the reviewer for the thorough and insightful comments, which have greatly helped improve the manuscript. We have addressed each point as follows:
Reviewer 2
Comments and Suggestions for Authors
Major Issues:
1. Inconsistent Outcome Measures Between Cohorts
The retrospective cohort uses positive percent agreement (PPA), negative percent agreement (NPA), and Cohen’s kappa, while the prospective cohort uses sensitivity, specificity, and the invalid result rate. This inconsistency makes it difficult to directly compare the performance of the ConfiSign HIV Self-Test across the two cohorts. The authors should justify the use of different outcome measures or use the same set of indicators for both cohorts to ensure comparability.
- Thank you for your insightful comment. To ensure comparability, we have recalculated and reported both sensitivity/specificity and PPA/NPA/PPV/NPV for each cohort where applicable. This allows for a direct comparison of diagnostic performance across retrospective and prospective analyses.
In line 204-208: The ConfiSign HIV Self-Test correctly identified all 400 HIV-positive samples, yielding a PPA of 100.0% (95% CI: 99.1%–100.0%). Of the 1,000 negative specimens, 8 (0.8%) were incorrectly identified as positive resulting in NPA of 99.2% (95% CI: 98.4%–99.7%) (Table 1). Corresponding sensitivity, specificity, PPV, and NPV were 100.0% (95% CI: 99.0%–100.0%), 99.2% (95% CI: 98.4%–99.7%), 98.0% (95% CI: 96.2%–99.0%), and 100.0% (95% CI: 99.6%–100.0%), respectively
- Lack of Inclusion and Exclusion Criteria
The authors do not specify the inclusion and exclusion criteria for participants in either the retrospective or prospective cohorts. This information is critical for understanding the generalizability of the study findings and the potential biases that may have been introduced. For example, were participants with certain comorbidities excluded? Were there any restrictions based on the time since HIV diagnosis? The authors should provide detailed inclusion and exclusion criteria for both cohorts.
- We agree that detailed inclusion and exclusion criteria are important for assessing the generalizability of the findings. We have now included a clear description of these criteria for both the prospective and retrospective cohorts in the revised manuscript.
In line 111-116: Blood specimens were included in the clinical trial if they were collected from participants who met all eligibility criteria, including confirmed HIV status by immunoassays, proper anonymization, and adherence to standardized collection and storage protocols. Specimens were excluded if they exhibited signs of microbial contamination, marked hemolysis or turbidity, improper storage or unverifiable storage conditions, were pooled from multiple sources, or had a volume less than 100 µL.
In line 124-139: Individuals aged 13 years or older who voluntarily agreed to participate and provided written informed consent were eligible for enrollment. All participants were required to be capable of independently collecting capillary blood samples and interpreting test results. For minors under 19 years of age, both the participant and their legal guardian received a full explanation of the study and co-signed the informed consent form. Although the study protocol allowed for the inclusion of participants aged 13 to 18 years with guardian consent, no minors were actually enrolled in the study. HIV-positive participants were individuals diagnosed with HIV/AIDS, as confirmed by medical records, who visited the study institutions. HIV-negative participants were individuals with no known risk factors or symptoms of HIV infection and who were unaware of their HIV status. A total of 112 HIV-positive individuals with confirmed diagnoses and 223 individuals with unknown HIV status, including healthy volunteers were recruited. Exclusion criteria included age under 13 years, failure to provide written informed consent, or psychological instability as determined by the principal investigator. Samples were also excluded if consent was withdrawn, if there was evidence of sample contamination or insufficient sample volume, or if documentation was incomplete.
- Insufficient Description of Statistical Methods
The statistical analysis section is too brief and lacks important details. For example, the authors mention calculating sensitivity, specificity, and Cohen’s kappa but do not describe how confidence intervals were computed or which statistical software was used. They also do not explain how sample sizes were determined for the retrospective and prospective studies. A more comprehensive description of the statistical methods is needed to allow readers to assess the validity of the results.
- We agree that the statistical analysis section required more detail, and we have revised the manuscript accordingly.
In line 184-199: Diagnostic performance was assessed by calculating sensitivity, specificity, agree-ment rate, Cohen’s kappa coefficient, positive percent agreement (PPA) and negative per-cent agreement (NPA), each with corresponding 95% confidence intervals (CI). PPA and NPA were calculated by comparing the results of the ConfiSign HIV Self-Test with those obtained from immunoassays used as reference standards. Confidence intervals for pro-portions were calculated using the Clopper-Pearson (exact) method. Cohen’s kappa coef-ficient was calculated to assess the level of agreement beyond chance. All statistical anal-yses were performed using MedCalc® Statistical Software version 23.2.8 (MedCalc Soft-ware Ltd, Ostend, Belgium) and GraphPad Prism version 10.5.0 (GraphPad Software, San Diego, CA, USA). All statistical tests were two-tailed, with p-values < 0.05 were considered statistically significant. For the retrospective study, the sample size was based on the requirements set by the Ministry of Food and Drug Safety (MFDS) of Korea for approval of HIV immunoassay devices, which recommend 400 HIV-positive samples and 1,000 HIV-negative samples. For the prospective study, the sample size was determined based on the maximum number of eligible HIV-positive individuals who could be enrolled dur-ing the study period across two tertiary hospitals.
- Limited Discussion of Early Seroconversion Performance
While the authors acknowledge that the ConfiSign HIV Self-Test has lower sensitivity in early seroconversion samples compared to fourth-generation assays, they do not discuss the implications of this limitation in sufficient detail. How might this affect the test’s real-world effectiveness? What guidance should be given to users regarding the timing of testing after potential exposure? The discussion should address these questions to provide a more balanced interpretation of the findings.
- Thank you for this important comment. We have revised the discussion to more thoroughly address the implications of reduced sensitivity during early seroconversion. The following text has been added in line 294-305
“Although the ConfiSign HIV Self-Test demonstrated high overall diagnostic accuracy, its lower sensitivity in early seroconversion cases should be carefully considered. As an antibody only assay, it may yield false-negative results if used during the serological window period, when anti-HIV antibodies are not yet detectable in blood. This window period typically ranges from 2 to 12 weeks after potential exposure, depending on the individual’s immune response and test characteristics. Consequently, individuals who test negative but have had a recent high-risk exposure should be advised that a negative result does not exclude acute HIV infection. In such cases, repeat testing after the window period or the use of a fourth-generation antigen/antibody assay or nucleic acid amplification test is recommended to improve early detection. Clear guidance regarding the optimal timing of self-testing and the limitations of antibody-only tests is essential to support informed decision-making and reduce the risk of missed early infections.”
- Generalizability of Subtype Inclusivity Findings
Although the test demonstrated broad reactivity across major HIV-1 subtypes and HIV-2, the sample size for certain rare subtypes and HIV-2 infections was limited. The authors should discuss whether the observed performance in these subtypes is sufficient to support global use of the test, particularly in regions where these subtypes are more common. Additional validation in diverse geographical settings may be necessary.
- Thank you for this insightful comment. We have addressed this point in the revised discussion as follows in line 308-315
“While the ConfiSign HIV Self-Test demonstrated broad reactivity across major HIV-1 subtypes and HIV-2, our study included only a limited number of samples from certain rare subtypes and from HIV-2 infections. Although our preliminary findings suggest that the test is capable of detecting these variants, the small sample sizes do not allow for definitive conclusions regarding its performance in these groups. Therefore, additional validation studies in regions where these subtypes are more prevalent—such as West and Central Africa—are warranted. Such studies are essential to ensure the test’s diagnostic reliability and global applicability across diverse epidemiological settings.
Minor Issues:
1. Clarity of Language
The English language needs further improvement for clarity and precision. For example:
Line 105-106: “...were originally submitted for routine HIV screening or diagnostic testing and analyzed using...” Here, “submitted” might be better replaced with “collected” or “obtained” to more accurately describe the process of sample acquisition.
- According to the reviewer`s comment, we have replaced “submitted” with “obtained” to more accurately describe the process of sample acquisition.
Line 116: “The prospective study enrolled consenting adults between October 2024 and December 2024...” It would be clearer to specify the exact dates (e.g., “from October 1, 2024, to December 31, 2024”).
- According to the reviewer`s comment, we have clarified the study period as “from October 1, 2024, to December 31, 2024”.
Line 121: “The target sample size included 112 HIV-positive individuals and 223 healthy volunteers with unknown HIV status.” The term “healthy volunteers with unknown HIV status” is contradictory. It would be better to state “223 individuals with unknown HIV status, including healthy volunteers.”
- According to the reviewer`s comment, we corrected as “223 individuals with unknown HIV status, including healthy volunteers.”
- Terminology Consistency
The authors should ensure consistent use of terminology throughout the manuscript. For example, they use both “ConfiSign HIV Self-Test” and “Confisign HIV Self-Test” (e.g., Line 205). A single consistent spelling should be used.
- We have reviewed the manuscript and ensured consistent use of terminology, specifically standardizing the spelling of “ConfiSign HIV Self-Test” throughout.
- Presentation of Results
The presentation of results in tables and figures could be improved. For example, in Table 1, the confidence intervals for PPA and NPA in the prospective study are not formatted consistently (e.g., “96.76%–100%” vs. “98.36%–100%”). Additionally, the figure caption for Figure 1 does not clearly explain what the S/Co ratio represents and how it was calculated. A more detailed caption would enhance understanding.
- Thank you for your constructive suggestions regarding the presentation of results. We have revised Table 1 to ensure consistent formatting of the confidence intervals (CIs) for both the retrospective and prospective cohorts. In addition, we have updated the caption for Figure 1 to provide a more detailed explanation of the signal-to-cutoff (S/Co) values.
- Missing References
Some statements lack appropriate references. For example, in the introduction (Line 56), the authors mention that “AIDS-related illnesses still claimed around 630,000 lives globally in 2023” but do not provide a reference for this statistic. All factual statements should be supported by references.
- According to the reviewer`s comment, We have added the appropriate reference for the global HIV/AIDS statistics in the introduction as below.
“HIV/AIDS continues to be a significant global public health concern, with approximately 39.9 million people living with HIV worldwide by the end of 2023 [1]. Despite progress in treatment and prevention, 1.3 million people acquired new HIV infections during 2023, and an estimated 630,000 people died from AIDS-related causes [1].” in line 53-56.
- Formatting of Abbreviations
The list of abbreviations should be checked for consistency and completeness. For example, “HIV” and “AIDS” are defined, but terms like “PPA” and “NPA” are not included in the abbreviations section, even though they are used extensively in the text. Including all non-standard abbreviations in the list would improve readability.
- Thank you for your helpful suggestion. We have reviewed and updated the list of abbreviations to ensure that all non-standard abbreviations, including “PPA” and “NPA”.
Reviewer 3 Report
Comments and Suggestions for Authors
The research topic is interesting and contemporary. Home self-testing for HIV is a topic of discussion not only in academic circles, but also among HIV risk groups and patient organizations.
I think that the study design is not as clean as possible from the influence of confounding factors. For example, the number of samples in the four groups included in the study design was not comparable. Characteristics of the patients studied are not presented.
It is not clear what proportion of the included patients with negative tests are from at-risk groups for HIV infection. There is no data whether there are among the tested people who are on PrEP, which is important for later seroconversion, etc. I think these details are important and authors should include them.
Author Response
We sincerely thank the reviewer for the thorough and insightful comments, which have greatly helped improve the manuscript. We have addressed each point as follows:
I think that the study design is not as clean as possible from the influence of confounding factors. For example, the number of samples in the four groups included in the study design was not comparable. Characteristics of the patients studied are not presented.
- Thank you for your thoughtful comments and for highlighting important aspects of our study design. We acknowledge that the sample sizes across the four groups were not directly comparable. This was an intentional aspect of our design, reflecting the distinct objectives and practical constraints of the retrospective and prospective components. Specifically, the retrospective analysis was structured to meet regulatory requirements, which recommend 400 HIV-positive and 1,000 HIV-negative samples for diagnostic performance evaluation. In contrast, the prospective study aimed to assess real-world, unassisted self-testing and was limited by the number of eligible HIV-positive individuals available during the study period.
- In response to your suggestion, we have included a summary of the demographic characteristics (age and sex) of the prospective participants in the revised manuscript in line 216-217” Of whom 172 (51.3%) were male and 163 (48.7%) were female. The median age of participants was 37.0 years (interquartile range: 31.0–47.0 years)”.
- Unfortunately, more detailed clinical information such as comorbidities or behavioral risk factors was not collected in this study. We added it to the discussion section as a limitation of present study in line 363-334 as “more detailed clinical information such as comorbidities or behavioral risk factors was not collected in this study”.
It is not clear what proportion of the included patients with negative tests are from at-risk groups for HIV infection. There is no data whether there are among the tested people who are on PrEP, which is important for later seroconversion, etc. I think these details are important and authors should include them.
- We appreciate the reviewer’s insightful comment regarding the importance of characterizing the risk profile of HIV-negative participants and documenting PrEP usage. Unfortunately, information on participants’ HIV risk behaviors and PrEP usage were not collected in this study, which limits our ability to assess the representativeness of the HIV-negative cohort and to evaluate potential implications for seroconversion risk.
- We have added a statement in the discussion section to acknowledge this limitation and have recommended that future studies collect detailed behavioral and clinical data, including PrEP usage, to more accurately assess test performance in key populations at risk for HIV infection.
In line 353-361: “Fourth, a notable limitation is the lack of detailed information on HIV risk profiles and pre-exposure prophylaxis (PrEP) use among participants in the HIV-negative cohort. As data on behavioral risk factors and PrEP status were not collected, it is difficult to determine the proportion of individuals at higher risk of HIV acquisition or to assess test performance among those on PrEP, who may experience delayed seroconversion. Future studies should incorporate risk stratification and PrEP usage data to enable more accurate and population-specific assessments of HIV self-test performance.”
Round 2
Reviewer 2 Report
Comments and Suggestions for Authors
The authors have addressed all my concerns.
Reviewer 3 Report
Comments and Suggestions for Authors
The authors of the manuscript have taken into account my comments and made the necessary corrections. I have no additional comments.